# Carrageenan-Based Acyclovir Mucoadhesive Vaginal Tablets for Prevention of Genital Herpes

**DOI:** 10.3390/md18050249

**Published:** 2020-05-11

**Authors:** Edisson-Mauricio Pacheco-Quito, Roberto Ruiz-Caro, Juan Rubio, Aitana Tamayo, María-Dolores Veiga

**Affiliations:** 1Department of Pharmaceutics and Food Technology, Faculty of Pharmacy, Complutense University of Madrid, 28040 Madrid, Spain; edissonp@ucm.es (E.-M.P.-Q.); mdveiga@ucm.es (M.-D.V.); 2Institute of Ceramics and Glass, Spanish National Research Council, CSIC, 28049 Madrid, Spain; jrubio@icv.csic.es (J.R.); aitanath@icv.csic.es (A.T.)

**Keywords:** genital herpes, acyclovir controlled release, mucoadhesive vaginal tablets, marine polymers, iota-carrageenan, hydroxypropyl methylcellulose

## Abstract

Women are the most affected by genital herpes, which is one of the most common sexually transmitted infections, affecting more than 400 million people worldwide. The application of vaginal microbicides could provide a safe method of protection. Acyclovir is a safe and effective medication for vaginal administration, and numerous benefits have been observed in the treatment of primary or recurrent lesions due to genital herpes. Vaginal tablets based on a combination of the polymers iota-carrageenan and hydroxypropyl methylcellulose were developed for the controlled release of acyclovir. Swelling, mucoadhesion and drug release studies were carried out in simulated vaginal fluid. The tablets, containing a combination of iota-carrageenan and hydroxypropyl methylcellulose, have an adequate uptake of the medium that allows them to develop the precise consistency and volume of gel for the controlled release of acyclovir. Its high mucoadhesive capacity also allows the formulation to remain in the vaginal area long enough to ensure the complete release of acyclovir. These promising formulations for the prevention of genital herpes deserve further evaluation.

## 1. Introduction

Sexually transmitted infections (STIs) are among the most common communicable conditions and affect the health and lives of people worldwide. The World Health Organization (WHO) estimates there are more than 1 million new curable STIs every day [1]. Genital herpes is one of the most widespread sexually transmitted infections in the world. Herpes simplex virus 2 (HSV-2) is the main cause of this disease [2]. It has been estimated that around 400 million people are infected with HSV-2 worldwide, and that approximately 20 million new cases occur each year [3]. More women than men are infected with HSV-2, as sexual transmission of HSV is more efficient from men to women than vice versa. It is a global threat to public health [4]. HSV-2 and Human Immunodeficiency Virus (HIV) have been shown to influence each other; HSV-2 infection increases the risk of acquiring a new HIV infection by approximately three-fold [5,6], and up to eight-fold if the exposure occurs soon after acquiring the HSV-2 infection [7,8].

Although there are definite therapies for the treatment of viral infections such as HSV-2, there is no cure for this disease. A viable strategy today is, therefore, to develop effective methods to prevent the spread of STIs, of which one of the most promising is the use of microbicides. Microbicides are chemical agents used topically by women within the vagina to prevent the spread of STIs, including HIV. The most commonly used vaginal dosage forms include gels, tablets and vaginal rings [9]. There is currently no effective microbicide, and many vaginal microbicide formulations do not cause the desired response due to certain limitations such as leakage, messiness, relatively low residence time, low dose accuracy and low stability, giving rise to an inadequate formulation and hence a lack of effectiveness. Additionally, STI prevention techniques do not reach all women, especially in developing countries, where the prevalence of sexually transmitted infections is high [4,10]. The development of a vaginal microbicide to effectively prevent the sexual acquisition of viruses and reduce new cases of HSV infection in women could be a safe method of protection for women and their partners. Unlike male or female condoms, microbicides are a potential preventive option that can be easily controlled by women themselves, and do not require the cooperation, consent or even the knowledge of their partner [11,12,13].

Acyclovir (ACV), a synthetic purine nucleoside analogue derived from guanine, was the first antiviral drug to specifically target a viral enzyme, DNA polymerase, to inhibit DNA chain elongation, and is one of the most effective and selective antiviral drugs. ACV has an antiviral effect on the herpes simplex virus-1 (HSV-1), herpes simplex virus-2 (HSV-2) and varicella zoster virus (VZV) by interfering with DNA synthesis and inhibiting viral replication [14]. ACV is a safe and effective drug for vaginal administration, used in the treatment of primary or recurrent genital herpes lesions. A clinical study by Corey et al. showed that topical acyclovir shortens the duration of viral shedding and accelerates healing of some genital herpes simplex virus infections [15,16,17], as well as preventing transmission of genital herpes. Different dosage forms containing ACV have been evaluated, including tablets [15,18], gels [19], intravaginal rings [20], microporous matrices [21] and nanoparticles [22].

Tablets offer many advantages, such as portability, precise dosing, high stability, ease of storage, handling and administration and feasibility of large-scale production at a low cost compared to semi-solid systems. Semi-solid systems have several limitations such as leakage, discomfort, complicated application and low residence time, and they may provide unsuitable doses due to the heterogeneous distribution [12,23]. Some reported solutions, such as liposomal delivery systems for vaginal administration of ACV, apart from the previously indicated disadvantages, have a low percentage of drug encapsulation efficiency and low stability of the liposomes in simulated vaginal fluid (SVF), due to the components of the acidic medium that induce a fast drug release. Thus limiting their application [24,25]. On the contrary, tablets could therefore be useful for the controlled release of ACV using polymers of natural or semi-synthetic origin [26]. The main advantage of these polymers is their biocompatibility, biodegradability and non-toxicity [27]. 

There is currently a widespread trend to seek raw materials in marine resources that can be successfully incorporated into biomedical applications. These resources include algae, crustaceans and other microorganisms, all of which provide compounds called marine polymers. Marine polymers are divided into three important groups, polysaccharides, proteins and lipids, and can be used for biomedical applications, such as in regenerative medicine and as delivery vehicles for controlled/sustained drug release, as well as for tailored biomaterials [28,29]. 

The priority focus in the study of biopolymers is on marine polysaccharides, which are important due to their sources and their ease of acquisition as a renewable resource. Seaweeds are the main source of marine polysaccharides such as alginates, fucoidans, laminarin, agar, carrageenans, galactans and ulvan [30]. Carrageenan, agar and alginate are the marine origin polysaccharides most widely used commercially [31]. 

Carrageenans are a group of linear sulphated polysaccharides present in the cell structure of *Rhodophyceae* algae. All carrageenans have a high molecular mass and are a linear sulphated polysaccharide of D-galactose and 3,6-anhydro-D-galactose (3,6-AG) linked by a-1,3 and b-1,4-glycosidic bonds, commonly used in the pharmaceutical industry for the development of hydrogels due to their high viscosity, gelling ability and stabilizing properties [32,33,34]. They can be classified into three main types: *kappa*, *iota* and *lambda*, depending on the number and position of the sulphate groups. The content of 3,6-anhydro-D-galactose (3,6-AG) determines the characteristics of the different types of carrageenan, and high levels of ester sulphate imply a lower gelling force and a lower solidification temperature [32,35]. Iota-carrageenan (iota-CG) contains 28% to 35% ester sulphate and 25% to 30% 3,6-AG units, and forms colloids and gels in aqueous media at very low concentrations. These gels are transparent and thermoreversible, and can be either elastic and cohesive or firm and fragile, depending on the combination of fractions used [33,34,35,36]. Carrageenans have been used in several studies, with effective results in vaginal administration, such as the vaginal gels containing carrageenan studied for the inhibition of the human papillomavirus (HPV), which showed that carrageenan has activity against HPV [37,38,39]. Another study demonstrated the advantage of combining carrageenan with Griffithsin, which acts as a broad-spectrum microbicide against HSV-2 and HPV in vitro and in vivo [40,41]. There is also evidence that carrageenan-based gel offers protection against HSV-2 transmission by binding to herpes virus receptors [42,43,44]. A number of reports focus on adding carrageenan to other polymers to leverage the gelling properties of carrageenan and achieve a good controlled-release profile [15,45]. The addition of carrageenan significantly decreased the erosion rate of a poloxamer 407-based gel, improved the sustained-release properties of acyclovir, and exhibited a synergistic bioadhesive effect with Carbopol^®^ in vivo [45].

Hydroxypropyl methylcellulose (HPMC) is a cellulose derivative that is widely included in hydrophilic matrix applications due to its stability, independent pH performance, worldwide regulatory acceptance and its versatility in the application of various pharmaceutical forms for controlled release [13,46,47,48]. Tablets formulated only with HPMC have shown an ex vivo mucoadhesion time of over 70 h and a sustained and complete drug release of over 40 h, confirming its applicability in the vaginal environment [49]. Hydroxypropyl methylcellulose has been incorporated into vaginal formulations such as gels [50], films [47,51] and tablets [15,52], exhibiting good mucoadhesion and a controlled release of antimicrobial drugs such as ACV [53], tenofovir [13], fluconazole [54] and clotrimazole [55]. 

Against this background, and since women are more affected by HSV—as sexual transmission of the virus is easier from men to women than vice versa—the aim of this study was to develop vaginal tablets loaded with ACV based on a combination of iota-CG, a polymer of marine origin with proven antiviral capacity, and HPMC, a semi-synthetic polymer, in order to obtain mucoadhesive tablets with ACV controlled release to protect women from the high incidence of HSV, and thus, comply with WHO guidelines [4].

## 2. Results and Discussion

### 2.1. Preparation of the Tablets

In the present study, eight batches of tablets with varying concentration of polymer (iota-CG, HPMC) and magnesium stearate were prepared, each tablet weighing 328 mg (drug included) and blank batches (without drug) of 228 mg weight. The composition of each batch (mg/tablet) is shown in Table 1. The tablets were cylindrical in shape (with an average diameter of 13 mm) and their height was directly weight-related; the heights of the tablets were 1.8–2.0 mm.

### 2.2. Infrared Spectroscopy

The spectroscopic characterization of the raw materials, the blank tablets and tablets with ACV by Attenuated total reflection Fourier transform infrared (ATR-FTIR) spectroscopy is shown in Figure 1. 

The fingerprint region of the typical spectra of the raw materials is shown in Figure 1A. The study of iota-CG by FTIR spectroscopy reveals the presence of very strong absorption bands in the 1210–1220 cm^−1^ region for the ester sulphate group, and the symmetric vibration of the two sulfonic groups at 847 and 801 cm^−1^. The glycosidic linkage occurs in the 1010–1080 cm^–1^ region and below, at about 947 cm^−1^, the vibration of the C-O-C bond in the 3,6-anhydro-D-galactose [56]. The IR spectrum of HPMC indicates the characteristics peaks of cellulose C-O-C at 1060 and 958 cm^−1^ [57]. 

The ATR-FTIR analysis of the blank tablets (Figure 1B) confirmed that the polymers and magnesium stearate (MgSt) do not interact with each other during compression, as polymer-polymer and polymer-MgSt are compatible, i.e., there are no signs of degradation of any of the components. However, there is a notable decrease in the band centered at 801 cm^−1^, corresponding to the sulfonic group in the tablet formulated with iota-CG (marked with an asterisk). No significant modifications are seen in the spectra on the tablet formulated with HPMC and the raw material itself or between HPMC and iota-CG when the two polymers are present.

The incorporation of ACV into the tablets (Figure 1C) maintains the fingerprint region of the ACV pure material, and the most characteristic bands correspond to the C=O bonds and are centered at 1709 cm^−1^; the bands corresponding to the amino groups occur at 1630 cm^−1^ and 1609 cm^−1^ [58]. The prepared batches—ACV combined with iota-CG (ACV-I), ACV combined with HPMC (ACV-H) and ACV combined with iota-CG and HPMC (ACV-IH1 and ACV-IH2)—showed the multiple bands of ACV and the bands corresponding to the tablets with no drug incorporated, indicating the complete compatibility of the drug with the polymers and MgSt.

### 2.3. Swelling Tests 

The numerical data on weight gain was used to calculate the degrees of swelling according to the formula described in the methodology section. Figure 2 shows the swelling/erosion profiles of each batch assayed when evaluating the change in weight that occurs in the medium.

Each positive value indicates that the percentage of swelling or weight gain is greater than the initial weight of the dry matrix, while each negative swelling ratio (SR) value indicates that the weight of the swollen matrix is less than the weight of the dry matrix due to the erosion or dissolution of the system in the medium. 

The swelling behavior of I tends to be very fast, since it reaches the maximum value in the first six hours, and erodes rapidly in acidic medium due to the nature of the polymer, which forms a fluid gel thanks to the presence of the divalent ion of the SVF (Ca^++^) [59]. Calcium is able to form intra-molecular bridges between the sulphate groups of adjacent anhydro-D-galactose and D-galactose residues of iota-CG [60], as can be seen in the photos in Figure 3.

H has a slow swelling behavior, and two characteristic zones on this tablet can be distinguished from the first hours of the test: a gelled outer layer (gummy phase) and an internal area in the vitreous state (non-hydrated core). The non-hydrated core allows slow swelling as the gel continues to erode, thanks to the gelled outer layer that protects the core from rapid swelling [49]. The nucleus remains in the formulation until about 120 h, as shown in Figure 3. 

Figure 2A also shows the swelling/erosion profiles of all the batches developed in mixtures of iota-CG with HPMC. The swelling behavior in batches of tablets containing a mixture of both polymers is conditioned by the nature of each polymer and the possible interaction between them. It can clearly be seen that the IH1 and IH2 tablets have the highest swelling rate. This is because the HPMC chains can swell completely when iota-CG is combined with HPMC, increasing the speed at which they form a gel and avoiding the permanence of the non-hydrated core observed in the H tablets, given the consistency of the fluid gel that originates I in aqueous medium, as confirmed in the sequence of photos in Figure 3. In both cases, the combination of polymers does not allow the constant presence of a core for more than 24 h, since after that point, it is impossible to distinguish this structure due to the higher swelling rate (Figure 3).

Figure 2B shows the swelling/erosion profiles of tablets with 100 mg of ACV. The swelling/erosion profiles are lower than in the blank batches in all cases, demonstrating the influence of the drug on the swelling behavior, which reduces the uptake of aqueous medium due to its limited solubility in water [24]. ACV-I has a lower swelling profile than I; the structure also tends to erode rapidly and is completely eroded after 48 h, as previously mentioned in I, although in these tablets, the presence of the core was observed until 24 h (Figure 3). The characteristic non-hydrated core of this polymer can be seen in ACV-H, and when ACV is incorporated in the tablet the nucleus can be observed for up to 144 h, as its presence further reduces the uptake of the medium. ACV-IH1 and ACV-IH2 showed less swelling than their blank batches, as seen in the photographs, which show two clearly defined areas in the tablets: an outer layer and a non-hydrated core. This indicates that the behavior of these batches is very similar to the swelling behavior of H. The presence of ACV clearly allows the formation of this area that reduces the uptake of the aqueous medium (Figure 3).

These results show that the combination of polymers enhances the swelling behavior when iota-CG is combined with HPMC, as it improves the swelling rate of iota-CG. The area under the curve (AUC) also reveals a difference between batches with or without ACV; and in all cases the SR_max_ and the AUC of the blank batches were greater, confirming that the presence of ACV modifies the erosion process. Finally, the t_max_ vary from 48–72 hours in the batches with or without drug, with the exception of I and ACV-I, whose t_max_ was only six hours in both cases, corroborating the rapid degradation of the formulation.

### 2.4. Microstructure of Swelling Witnesses

The swelling witnesses were analyzed by scanning electron microscopy (SEM), where the lost water was substituted by pores in a lyophilization process [61]. As can be seen from the micrographs in Figure 4, the witness microstructures vary considerably depending on the nature of the polymer. 

The micrograph of the swelling control of I (Figure 4A) reveals the formation of a homogeneous structure with small pores. In contrast, the structure of the swelling control of H has pores in a channel form (Figure 4B), which clearly differentiates it from the structure of control I. The combination of iota-CG and HPMC (batches IH1 and IH2) produces very similar structures to H. This is in line with the swelling/erosion profiles described above, which were similar but not superimposable. That is, when the batches formed by two polymers swell, the ordination of the HPMC chains predominates over that of the iota-CG chains, which also corresponds to the formation of a fluid gel from iota-CG compared to the more viscous gel developed by HPMC, and which is reflected in the numerical values of SR.

The incorporation of ACV (Figure 4E–H) did not modify the structures originated by the polymers formulated alone (ACV-I, ACV-H) or in combination (ACV-IH1, ACV-IH2), since the ACV particles do not gel and consequently do not modify the arrangement of polymer chains when they swell.

Control I (Figure 4A) has a uniform structure, with round pores that are partially distorted after the incorporation of ACV (Figure 4E). This characteristic microstructure allows the formulation to erode easily, as was deduced from its swelling behavior, which did not reach 48 h in the testing media. The microstructure of the batches formulated solely with HPMC can be described as elongated channels (Figure 4B) with small pores in their walls in the presence of ACV (Figure 4F), allowing a gradual uptake of the surrounding medium which is translated into a moderate swelling behavior compared to the other batches. Moreover, as mentioned in previous sections, the presence of the channels allows the non-hydrated core to swell slowly as the gel progressively erodes. Water mobility plays a role in controlling the swelling and erosion of the tablets [13]. The HPMC has a gradual swelling, starting in the outer layers until it arrives at the nucleus, then progressively hydrates when the external gel has been eroded and the water reaches the nucleus [13,62]. 

The most dramatic change in the microstructure occurred in cases where ACV was introduced in the batches with HPMC and iota-CG (Figure 4C,D,G,H). The characteristic channels of the HPMC structure appear to be interconnected by means of thin branches that resemble the microstructure of the iota-CG, a characteristic that is more pronounced when the drug molecule is incorporated (Figure 4G). In Figure 4D,H, these branches are less noticeable due to the lower proportion of this polymer. The influence of HPMC determines the swelling behavior of these formulations (high swelling), as when iota-CG is mixed with HPMC, they form a microstructure similar to that of H, which allows a greater uptake of the medium. In this case, the tablets do not have the characteristic non-hydrated core seen in the tablets with H, but form a structure that evenly captures the medium.

The Figure 5 shows the results obtained from Hg intrusion porosimetry on the swelling witnesses of all the batches.

The PSDs obtained from the Hg intrusion curves reveal different pore sizes and volumes depending on the polymer used. The tablets containing solely the iota-GC polymer (shown in Figure 5A) have the minimum pore size values (about 40 µm), whereas the formulations containing the polymer HPMC also reveal a bimodal distribution with pores of about 100 µm and 50 µm (these latter to a lesser degree). The combination of the two polymers produces a monomodal distribution with an intermediate pore size between the two individual polymers. The pore size determines its erosion in acidic medium, thus, the gel is no less consistent and crumbles earlier, according to the data studied in the swelling test.

When ACV is incorporated into the formulation, the PSDs in Figure 5B differ significantly from the tablets with no drug included. Although the same trend is observed in the mean pore size (mean PSD ACV-I < ACV-H), it is notable that the bimodal character of the PSD of ACV-H is maintained when the two polymers are combined, especially in the batch containing the minimum amount of iota-CG, i.e., ACV-IH2. However, instead of a mean PSD between the two polymers, the maximum pore volume occurs at about 100 µm. Except in the case of the ACV-I formulation, a decrease in pore volume can be seen with the incorporation of the ACV compared to the original tablet with no ACV included.

From these observations, it can be concluded that the amount and type of polymer influence the pore size and volume, as seen in the SEM micrographs. The pore volume is thus determined by the amount of iota-CG, since the higher the amount of this polymer, the higher the volume of the pore; in contrast, the pore volume decreases when there is more HPMC in the combined tablets, and the incorporation of ACV also strongly influences the porous properties in all cases. 

### 2.5. Drug Release

The ACV release profiles of all the formulations are shown in Figure 6. Sustained release of more than 24 hours occurs in all cases. 

ACV-I had a total release in 48 hours, and released 95% in the first 24 h, demonstrating that rapid erosion in the acid medium of this tablet produces greater drug release in a shorter time.

In contrast, ACV-H tablets exhibited the best control of ACV release, attaining the complete release of the drug at 144 h. This is clearly related to the swelling/erosion profiles of this formulation, which is intermediate compared to the other formulations. This formulation also has different zones—a gelled outer layer and an internal area in the vitreous state—which allow a prolonged release of ACV.

For batches ACV-IH1 and ACV-IH2, the total release of ACV occurs at 96 h, representing an intermediate release compared to the other formulations. These formulations control the release of the drug more effectively than ACV-I, thanks to the consistency and volume of the gel formed by this combination. This is supported by the swelling/erosion profiles, since these mixtures have the highest swelling rate of all the tablets, allowing a faster release compared to ACV-H. There are no significant differences in the release profiles of ACV-IH1 and ACV-IH2, regardless of the amount of HPMC or iota-CG. This confirms that the iota-CG/HPMC combination produces a mixed gel that can maintain its properties despite changes in the proportion of the two polymers, already described in other studies when HPMC is mixed with chitosan or *kappa*-carrageenan [13,15,63,64]. 

Based on the release studies, the formulations can therefore have three types of release: a quick release in the case of ACV-I; an intermediate release in batches with a mixture of polymers ACV-IH1 and ACV-IH2; and a prolonged release with ACV-H, as they offer a sustained release of more than 90% of the drug over 120 h. The polymer mixture allows the modulation of the characteristics of each component; thus, the formulation can be improved to allow controlled release profiles. The control of the drug release is due to the nature of the polymers, the pharmaceutical form and how they act in the acidic medium of the SVF.

The tablets containing the combination of polymers are the most suitable for our purpose, since, as they contain the two types of polymers, they allow a controlled release of the drug on the one hand, and—on the other—they keep the iota-CG in the formulation, which is very important due to its antiviral properties.

Mathematical models were used to determine the drug release mechanisms. After analyzing each sample, Korsmeyer-Peppas, Hixson-Crowell and Hopfenberg were found to be the models that best fit to the experimental results (Table 2).

In the case of the Korsmeyer-Peppas model, when the formulations are cylindrical tablets as in this research, a value of *n* ≤ 0.45 means that ACV release follows a pure diffusion (Fickian process), and a value of *n* between 0.45 and 0.89 indicates an anomalous transport (combination of simultaneous processes) where drug diffusion and relaxation of the polymer fibers occur simultaneously. When *n* takes a value equal to 0.89, drug release occurs through transport case II; if n > 0.89, drug release occurs through transport Supercase II. Both Case II and Supercase II involve the structural modification of the polymer matrix (relaxation of the polymer chains) [65,66]. ACV-I has a value of *n* > 0.89, meaning it follows a Supercase II drug release, implying extreme drug transport. The values of *K_KP_* constant (Table 2) of the ACV-I batch have a much higher value than the other batches that fit this kinetic, suggesting that ACV release occurs directly through the relaxation of the polymer chains. This was corroborated with the swelling test, where the formulation forms a fluid gel that does not allow the prolonged release of the drug for more than 48 h, due to the behavior of iota-CG in the acidic medium.

Batches ACV-H, ACV-IH1 and ACV-IH2 can be fitted to Korsmeyer-Peppas with a good correlation. In these batches, the *n* value is between 0.45 and 0.89 in all cases, suggesting that the release of ACV is due to simultaneous processes, namely the relaxation of the polymer chains and Fickian diffusion process. The release of ACV occurs because the polymers swell and the ACV diffuses through the resulting gel. This was confirmed by the data obtained in the swelling test, where batches ACV-IH1 and ACV-IH2 formed a structured gel with the highest swelling rate of all the batches studied; unlike ACV-H, which forms a long-lasting gel with an intermediate swelling rate.

ACV-I is the batch with the poorest fit to the Hixson-Crowell model, since it does not maintain its structure and erodes rapidly, as mentioned in the swelling test. ACV-H, ACV-IH1 and ACHI2 can fit this kinetics, as in the swelling test, where the structures maintain their shape and their size decreases over time due to the presence of HPMC, which structures these formulations more effectively.

Finally, the Hopfenberg model confirms that the process of ACV release occurs through erosion, as shown in Table 2. The correlation coefficients are high in all cases and are related to the Korsmeyer-Peppas correlation coefficients.

These results indicate that the formulations have simultaneous release processes, including the relaxation of the polymer chains, erosion and diffusion of ACV through the gel.

Similarity factor (*f_2_*) values were used to compare the experimental results (Table 3). 

Data were compared from batches with the single polymer and with the polymer combinations in different ratios. The comparison of batch ACV-I with batches ACV-H, ACV-IH1 and ACV-IH2 showed no similarity in any case; as already mentioned, this is due to the rapid release of ACV in ACV-I. Additionally, according to f2 values, there is no similarity either between batches ACV-H, ACV-IH1 and ACV-IH2, as the batches with the polymer mixture had an intermediate release in both formulations.

In contrast, ACV-IH1 and ACV-IH2 are similar, confirming that there are no significant differences, regardless of the proportions of polymers in which they are combined.

### 2.6. Mucoadhesion Assessment

Figure 7 shows the results of the mucoadhesion forces and work of mucoadhesion for all the batches. It can be seen that all the formulations can attach to the vaginal mucosa, and that a value of between 0.16 N and 0.29 N is required to separate the tablets. This confirms the mucoadhesiveness of all the polymers assayed, thanks to their adhesion mechanisms mediated by hydrogen bonds and electrostatic interactions [67,68]. 

It can clearly be seen that the formulations with one polymer (ACV-I and ACV-H) have higher force and work of mucoadhesion values, and that these values decrease when the formulations contain a combination of polymers (ACV-IH1 and ACV-IH2). This was also observed in an earlier study where polymers were tested alone and in mixtures [15].

Once it was verified that all the formulations had mucoadhesive properties, the next step was to determine how long they remained bonded to the vaginal mucosa.

Figure 8 shows the residence times of all batches attached to the vaginal mucosa. ACV-I is the batch with the shortest residence time, with only 1 hour of adhesion. ACV-H shows a good initial adhesion to the vaginal mucosa and a residence time of 120 h. In contrast, the tablets with iota-CG/HPMC remain attached to the mucosa for prolonged periods of time, more than 140 hours in both cases. The higher the amount of iota-CG, the longer the mucoadhesion time (ACV-IH1 > ACV-IH2). This may be due to the interaction that occurs when these two polymers are mixed, allowing the hydrogen bonds of HPMC and the sulphate groups of iota-CG to bind with the mucosa.

In all cases, the formulation interacts with the vaginal mucosa and prolongs the residence time of the pharmaceutical dosage, thus reducing dose frequency [69]. Mucoadhesive formulations are precisely modified release dosage forms, which make it possible to control the place, time, duration or magnitude of their action [70,71,72,73]. 

We can report that the best results were obtained in the batches containing the combination of polymers ACV-IH1 and ACV-IH2, since their residence time in the mucosa allows the complete release of the drug. As their time of permanence exceeds the drug release, the formulation remains in contact with the patient for longer, thus ensuring that the drug is completely released. Of the two aforementioned formulations, ACV-IH1 would be a good candidate for future tests, as it contains a higher percentage of iota-CG, making it suitable as an antiviral adjuvant of the formulation.

Although ACV-H has a prolonged release that meets the needs of this research, the residence time is less than the release time, thus, this formulation is not suitable for use in subsequent studies (Figure 8). 

Another circumstance that must be considered is vaginal turnover, a physiological mechanism whereby foreign elements are removed from the vaginal epithelial environment, with a renewal time of 96 h [74]. This is a limitation for mucoadhesive systems; however, this would not be a problem in the case of ACV-IH1 and ACV-IH2, as these tablets release 100% of the drug in 96 h, thus complying with the proper dosage.

Finally, it was verified that the carrageenan alone does not have prolonged mucoadhesion times [15], which in our case does not allow its use in controlled release formulations. However, when iota-CG is associated with HPMC, the combination enhances the properties of each component and allows both prolonged mucoadhesion times and controlled release. 

These proposed vaginal tablets would offer women full protection against the high incidence of HSV worldwide, and especially in developing countries such as in sub-Saharan Africa which have the highest incidence in the world, thus conforming to the guidelines proposed by WHO. 

## 3. Materials and Methods 

### 3.1. Materials 

Acyclovir (ACV, lot: 701376) was obtained from Lab. Reig Jofré (Toledo, Spain). *Iota*-carrageenan (iota-CG, lot: SLBB2304V) was provided by Sigma-Aldrich (St. Louis, MO, USA). Hydroxypropyl methylcellulose—Methocel^®^ K 100M CR (HPMC, lot: DT352711), MW: 72 × 10^4^ g/mol, was kindly supplied by Colorcon Ltd. (Kent, UK). Magnesium stearate PRS-CODEX (MgSt; lot: 85269 ALP) was acquired from Panreac (Barcelona, Spain). All other reagents used in this study were of analytical grade and used without further purification. Demineralized water was used in all cases.

### 3.2. Methods 

#### 3.2.1. Preparation of Tablets and Infrared Spectroscopy

Vaginal tablets were developed from physical mixtures of iota-CG, HPMC and iota-CG/HPMC with MgSt. In all cases, each tablet contained 100 mg of ACV. The selection of this amount of drug is based on a common application in a commercialized vaginal formulation (topical ACV cream at 5% *w*/*w*). Blank tablets (without drug) were also prepared in order to evaluate the possible interactions between both polymers. The tablets were manufactured with a 13 mm diameter (ID) hardened steel dry pressing die set for pellet. Each physical mixture was compressed with a constant force of five tons for four minutes, and the batches were stored in a desiccator until further evaluation. The tablets were evaluated for weight variation using a precision balance (METTLER^®^ AT 200, Mettler-Toledo S.A.E., Barcelona, Spain). Tablets height and diameter were measured using a digital micrometer (INSIZE^®^ 3108-25A, Suzhou, China). 

Attenuated total reflection Fourier transform infrared (ATR-FTIR) spectroscopy was used to characterize the raw materials, blank tablets and tablets with ACV with a Perkin-Elmer spectrophotometer instrument equipped with a MIRacle™ accessory designed for ATR-FTIR measurements (Perkin-Elmer, Waltham, MA., USA) to evaluate the possible polymer-polymer, polymer-MgSt and polymer-ACV interactions before and after compression.

#### 3.2.2. Swelling Tests

Each batch was evaluated in simulated vaginal fluid (pH = 4.2) [75], using the method described by Ruiz-Caro et al. [76]. Each tablet was fixed to a stainless-steel disc, 30 mm in diameter, with a cyanoacrylate adhesive (Loctite^®^, Henkel, Austria). This preparation was then placed in a beaker containing 80 mL of SVF, then in a thermostatized shaking water bath (Selecta^®^ UNITRONIC320 OR, Barcelona, Spain) at 37 ± 0.1 °C and at 15 opm. At specific time intervals, the discs were removed from the medium, placed on filter paper to remove the excess liquid and weighed on a precision balance until the complete dissolution or erosion of the tablet. Each sample was tested in triplicate. The swelling ratio (%) for each sample was calculated according to the following Equation (1) [77]:(1)$$SR\left(\%\right)=\left(\frac{Ts-Td}{Td}\right)\begin{array}{cc} \times & 100 \end{array}\;,\;$$
where *T_s_* corresponds to the weight of the swollen tablet and *T_d_* to the weight of the dry tablet. 

At t = 6, 24, 48, 72, 96, 120, 144 and 168 h, photographs were taken with a digital camera (Canon^®^ EOS 100D 18.0 megapixels, Tokyo, Japan), to observe the aspect and evolution of the tablets in contact with SVF. 

#### 3.2.3. Preparation and Characterization of Swelling Witnesses. SEM Microscopy and Hg Porosimetry

In order to differentiate the internal structure adopted by the tablets when introduced in SVF, swelling witnesses were obtained according to the time in which each sample reaches the maximum SR, using the same method as in the swelling test. The tablets were fixed to a stainless-steel disc and immersed in SVF inside a beaker, which was then placed in the thermostatized shaking water bath (Selecta^®^ UNITRONIC320 OR, Barcelona, Spain) at 37 ± 0.1 °C and 15 opm. The tablets were maintained under these conditions until the maximum SR was reached, previously quantified in the swelling test. Each tablet was then extracted from the medium and lyophilized (Lio-Labor^®^; Telstar, Barcelona, Spain) to obtain the swelling witnesses. Each batch was analyzed by electron microscopy using a scanning electron microscope SEM (JEOL JSM 6400) at 20 KV, where the samples were previously prepared by evaporative coating with graphite (Q150T Turbo-Pumped Sputter Coater/Carbon Coater).

The pore size distributions (PSD) of the witnesses were determined by mercury porosimetry using an Autopore II 9215 (Micromeritics Corp., Norcross, GA, USA). The corresponding PSD data were calculated from the intrusion curves, assuming cylindrical pore shapes in all cases.

#### 3.2.4. Drug Release 

The release of ACV from the batches was evaluated with the method described by Sánchez-Sánchez et al. [15]. Each tablet was inserted in a borosilicate glass bottle containing 80 mL of SVF and placed in a thermostatized shaking water bath (Selecta^®^ UNITRONIC320 OR, Barcelona, Spain), with an experimental temperature of 37 ± 0.1 °C at 15 opm. The test was performed in triplicate. Samples of 5 mL were removed from each bottle at pre-established times and filtered to eliminate the particles in the suspension medium; the medium was replaced with the same volume of SVF at the same temperature. ACV concentrations in the SVF were quantified by UV-Vis spectroscopy at a wavelength of 251 nm in a Shimadzu^®^ UV-1700 spectrophotometer (Kyoto, Japan). 

In order to determine the process of dissolution of the drug from the tablets, the drug release experimental data were fitted to various model-dependent methods (Korsmeyer-Peppas, Hixson-Crowell and Hopfenberg models) to investigate the kinetics of drug release from the different batches [65,66,78,79].

According to Korsmeyer-Peppas, the drug release as a function of time follows Equation (2):(2)$$\frac{Q_{t}}{Q_{\infty }}=K_{KP}\times t^{n},$$
where *Q_t_/Q_∞_* is the fraction of drug released at time *t*, *K_KP_* is a constant incorporating the structural and geometric characteristics of the tablet and *n* is the release exponent [65]. This model defines the drug release mechanism according to the value of *n*. 

The mathematic model described by Hixson and Crowell can be applied to dosage forms, where drug release takes place in parallel planes and the dimension of the dosage form is reduced proportionally, although its shape remains constant [80]. It is expressed as Equation (3), where *Q_0_*, *Q_t_* and *K_HC_* are the initial amount of drug, the drug released at time *t* and a constant that includes the surface-volume relation, respectively:(3)$$Q_{0}^{1∕3}-Q_{t}^{1/3}=K_{HC}\times t,$$

The Hopfenberg model is applied to surface-eroding dosage forms and is expressed as Equation (4), where *Q_t_* is, again, the drug released at time *t* and *Q_0_* is the total drug contained. *K_HF_* is the Hopfenberg rate constant, which includes the expression *K_0_/C_0_a_0_*, where *K_0_* is the erosion rate constant, *Q_0_* is the initial drug concentration in the dosage form and *a_0_* is half the initial system thickness. *n_H_* is the Hopfenberg exponent, which is related to the geometry and has a value of 1 (for slab), 2 (for cylindrical) or 3 (for spherical) depending on the system:(4)$$\frac{Q_{t}}{Q_{0}}=1-{\left[1-k_{HF}\times t\right]}^{n_{H}},$$
For our tablets, this expression can be summarized in Equation (5):(5)$$1-\sqrt{Q_{0}-Q_{t}}=k_{HF}\times t,$$

The similarity between the dissolution profiles obtained from our tablets was compared using a model independent index described by Moore and Flanner to establish the effect of the polymers on ACV release [81]. This similarity factor (*f_2_*) was calculated according to Equation (6).
(6)$$f_{2}=50\times \log\left\{{\left[1+\left(\frac{1}{n}\right)\sum_{j=1}^{n}\;W_{j}{\left|R_{j}-T_{j}\right|}^{2}\right]}^{-0.5}\times 100\right\},$$
where *n* is the number of samples for each dissolution test, *R_j_* and *T_j_* is the drug release percentage at each time for the reference and test product respectively and *W_j_* is a weight factor (*W_j_* = 1 in this work). A value for *f_2_* > 65 indicates a similarity between profiles of over 95%, while *f_2_* < 65 denotes non-similar profiles [79]. 

#### 3.2.5. Mucoadhesion Assessment

To establish the tablets’ capacity to adhere to the vaginal mucosa throughout the administration time, the work and force necessary for detachment was assessed using the method described by Cazorla-Luna et al. [67]. The mucoadhesion test was assessed with the TA.XT*plus* Texture Analyzer (Stable Micro Systems, Godalming, UK). The dry tablets were fixed to a 20 mm stainless-steel probe with double-sided adhesive tape. Square fragments of 2 × 2 cm of bovine vaginal mucosa (obtained from a local slaughterhouse) were fixed to a petri dish with cyanoacrylate adhesive. The probe with the tablet was moved at a speed of 1 mm/s until it came into contact with the mucosa, applying a contact force of 500 g for 30 s. The probe was then separated from the mucosa at a speed of 0.1 mm/s until the complete detachment of the tablet. The force applied during the detachment of the formulation was measured at a rate of 500 pps. The force applied vs. the distance covered by the probe was measured, and the maximum force required to separate the tablet from the mucosa was recorded. 

An ex vivo mucoadhesion test was applied to determine how long the tablets remained adhered to the vaginal mucosa, according to the method described by Notario-Pérez et al. [13]. A sample of bovine vaginal mucosa was fixed to an 8.5 cm × 5 cm stainless steel plate with cyanoacrylate adhesive. Each tablet was then adhered to the mucosa, applying a pressure of 500 g for 30 s. The preparation was placed at an angle of 60° inside a beaker containing SVF, and then in the thermostatized shaking water bath (Selecta^®^ UNITRONIC320 OR, Barcelona, Spain) at 37 ± 0.1 °C and 15 opm. All batches were evaluated in duplicate, and the residence time of each batch was assessed by visual observation of the samples. 

## 4. Conclusions

Combinations of different types of polymers allow the development of robust and effective vaginal tablets capable of controlling the release of ACV and thus useful for preventing genital herpes.

The combination of a marine-origin polymer—iota-CG—with proven antiviral capacity and a semi-synthetic polymer—HPMC—in vaginal tablets achieves the complete and controlled release of acyclovir. Of all the formulations tested, the tablets containing iota-CG and HPMC are indicated for helping prevent the sexual transmission of genital herpes. The microstructure formed by the mixture of polymers allows an adequate swelling rate, appropriate residence time in the mucosa and controlled release of acyclovir, making this an effective application for the prevention of genital herpes in women, thus reducing the spread of HSV through sexual contact. 

## Figures and Tables

**Figure 1 marinedrugs-18-00249-f001:**
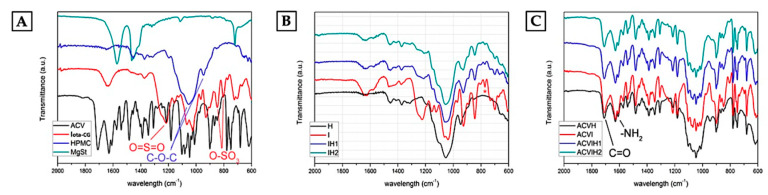
ATR-FTIR spectra of raw materials and tablets. **(A)** Raw materials: ACV, hydroxypropyl methylcellulose (HPMC), iota-carrageenan (iota-CG), magnesium stearate (MgSt); **(B)** blank tablets (H, I, IH1, IH2); and **(C)** tablets with ACV (ACV-H, ACV-I, ACV-IH1, ACV-IH2).

**Figure 2 marinedrugs-18-00249-f002:**
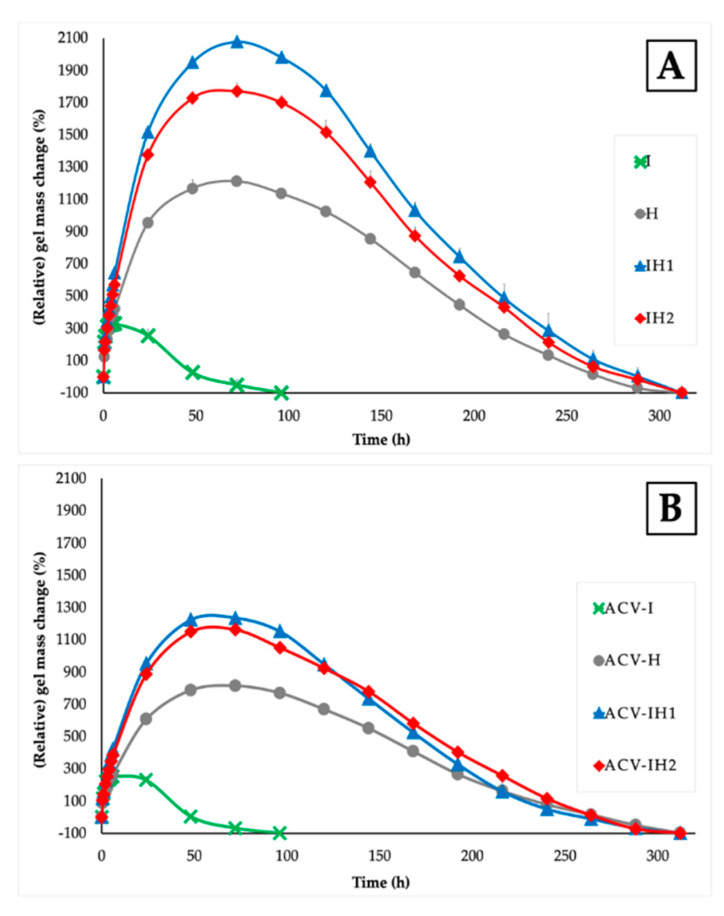
Swelling/erosion profiles obtained from (**A**) blank tablets and (**B**) tablets with ACV in SVF.

**Figure 3 marinedrugs-18-00249-f003:**
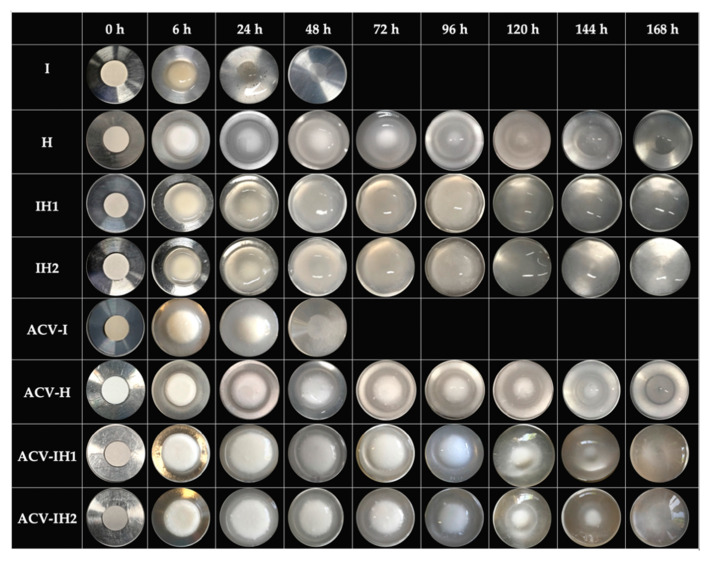
Swelling photos of tablets with and without ACV in SVF.

**Figure 4 marinedrugs-18-00249-f004:**
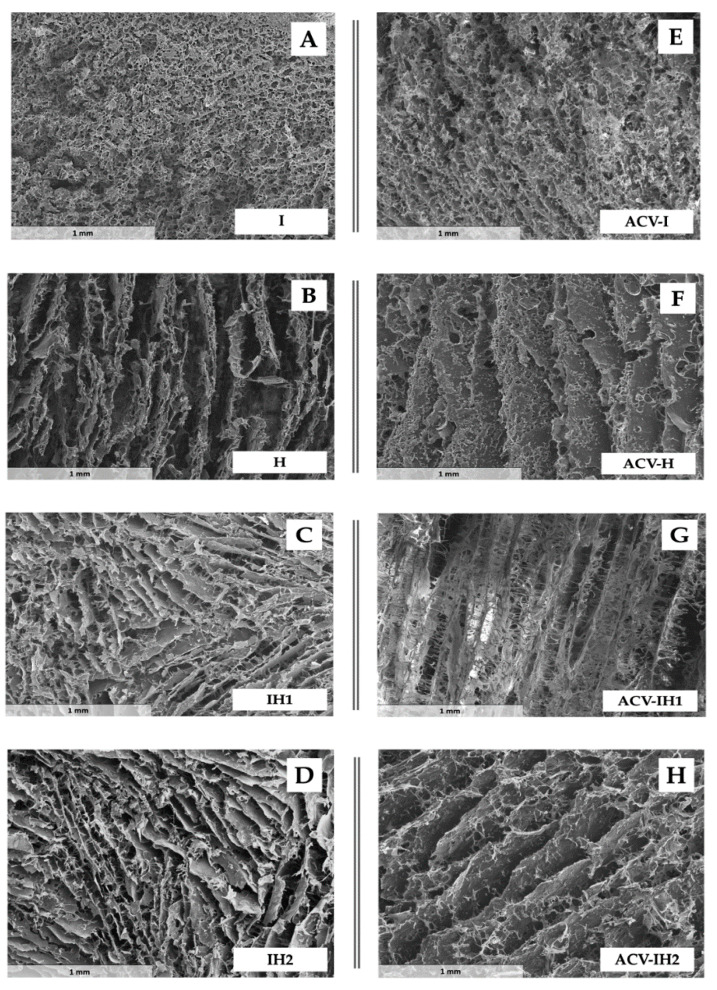
Electron microscopy micrographs of swelling witnesses of (**A**) I, (**B**) H, (**C**) IH1, (**D**) IH2, (**E)** ACV-I, (**F**) ACV-H, (**G**) ACV-IH1 and (**H**) ACV-IH2.

**Figure 5 marinedrugs-18-00249-f005:**
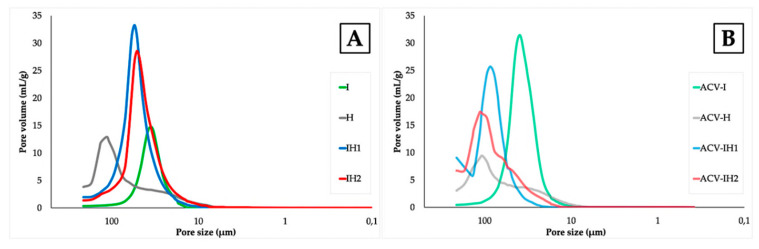
Results obtained from Hg porosimetry on the swelling witnesses of all the batches. (**A**) Blank tablets and (**B**) tablets with ACV.

**Figure 6 marinedrugs-18-00249-f006:**
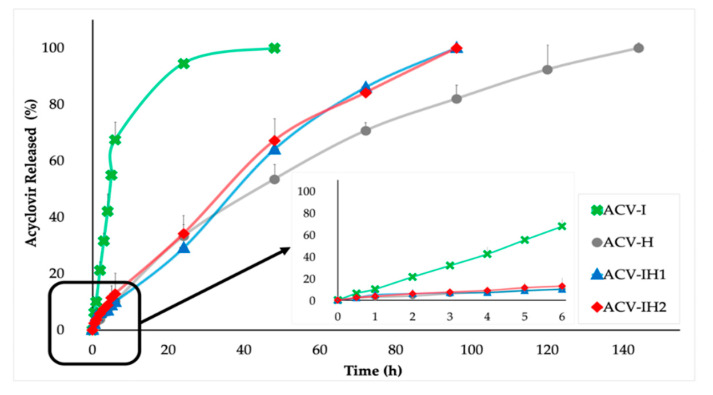
Acyclovir release profile in SVF from tablets based on iota-CG, HPMC and combinations.

**Figure 7 marinedrugs-18-00249-f007:**
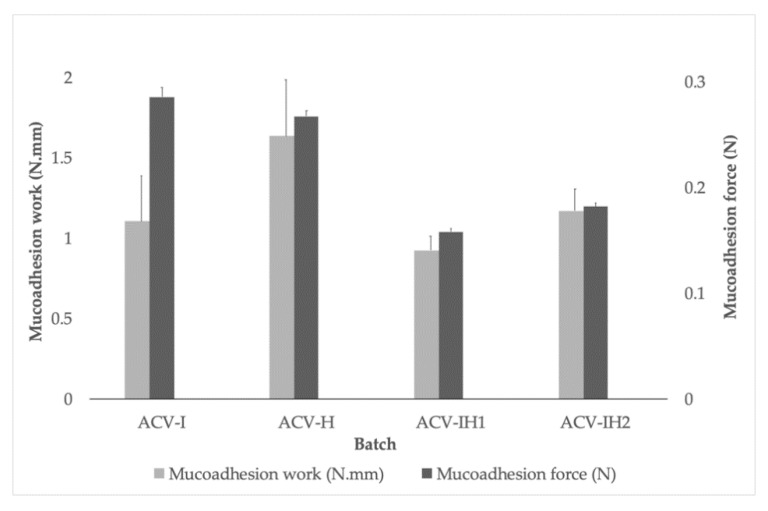
Work and force of mucoadhesion for the batches tested on bovine vaginal mucosa.

**Figure 8 marinedrugs-18-00249-f008:**
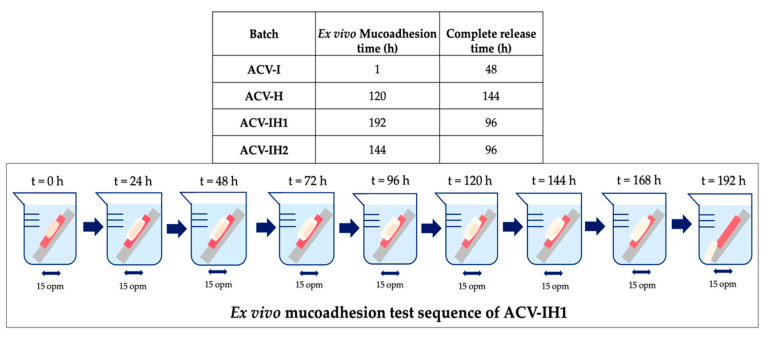
Summary of the data obtained from mucoadhesion residence time and drug release for each batch, and an ex vivo mucoadhesion test sequence of ACV-IH1.

**Table 1 marinedrugs-18-00249-t001:** Composition of the batches prepared in mg/tablet. ACV: Acyclovir.

Batch	*Iota*-Carrageenan	Hydroxypropyl Methylcellulose	Magnesium Stearate	ACV
I	225		3	
H		225	3	
IH1	135	90	3	
IH2	90	135	3	
ACV-I	225		3	100
ACV-H		225	3	100
ACV-IH1	135	90	3	100
ACV-IH2	90	135	3	100

**Table 2 marinedrugs-18-00249-t002:** Correlation coefficients obtained when experimental data are fitted to the different mathematical models and kinetic constants.

Batch	Korsmeyer-Peppas			Hixson-Crowell		Hopfenberg	
	R^2^	*K_KP_*	*n*	R^2^	*K_HC_*	R^2^	*K_HF_*
ACV-I	**0.9931**	0.1127	0.9633	0.9789	0.0513	**0.9917**	0.0655
ACV-H	**0.9900**	0.0279	0.7410	**0.9994**	0.0045	**0.9972**	0.0060
ACV-IH1	0.9830	0.0326	0.6474	**0.9860**	0.0063	**0.9919**	0.0083
ACV-IH2	**0.9960**	0.0368	0.6822	**0.9961**	0.0061	**0.9970**	0.0081

**Table 3 marinedrugs-18-00249-t003:** Similarity factor (*f_2_*) values for the release profiles. Comparisons with significant difference (*f_2_* < 65) are in bold.

Batch	ACV-I	ACV-H	ACV-IH1	ACV-IH2
**ACV-I**		**23.57**	**23.62**	**24.73**
**ACV-H**			**60.27**	**59.99**
**ACV-IH1**				79.11
**ACV-IH2**				**------**

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
