# Peer review of "Carrageenan-Based Acyclovir Mucoadhesive Vaginal Tablets for Prevention of Genital Herpes"

_marinedrugs, 2020, doi:10.3390/md18050249_

Round 1

Reviewer 1 Report

The aim of the peer-reviewed work is devoted to developing vaginal tablets loaded with Acyclovir based on a combination of natural polymer -carrageenan with proven antiviral capacity, and semi-synthetic polymer. The paper presented in a logical manner and is interesting as far as the subject matter is concerned. Although carrageenans, as I know, are widely used in the delivery systems of various drugs, including acyclovir, such systems are insufficiently explored and poorly studied. Therefore, this manuscript is a very important contribution to the development of mucoadhesive preparations with antiviral activity to protect women from the high incidence of HSV. The study is done very comprehensively with the application of several complementary techniques, from characterization of tablets of various compositions to the release of Acyclovir from them and interactions with mucin of solid polymers from animal tissue. I believe that this manuscript once published will become a key and very important reference regarding mucoadhesive characteristics of this important class system for controlled release drug. This is an interesting, complete and well designed paper, and I think, it deserves publication after a minor revision

  1. I believe that the results section should begin with the characteristics of the substances, in this case tablets. Please in the results section before Infrared spectroscopy give the designation of the obtained tablets and a brief description of their receipt. This will help to better perceive the results, since the designation of tablets is given only at the end of the article in the methods
  2. 2. Please transfer table 3 from the methods section to the results section.
  3. 3. Please show clearly in the infrared spectroscopy image a range from 1000 to 500 nm. It is know that IR spectra of sulfated carbohydrates display absorption bands in the region of 800-850 nm, and axial sulfate at position 2 of a 3,6- anhydrogalactose residue has a specific absoption band at 805 nm characterizated of iota carrageenan. This area is poorly shown in Figure 1. For quantitative evaluation, it is better to obtain the spectrum of adsorption, but not transmission. .

Author Response

Response to Reviewer 1 Comments

Reviewer’s comment: The aim of the peer-reviewed work is devoted to developing vaginal tablets loaded with Acyclovir based on a combination of natural polymer -carrageenan with proven antiviral capacity, and semi-synthetic polymer. The paper presented in a logical manner and is interesting as far as the subject matter is concerned. Although carrageenans, as I know, are widely used in the delivery systems of various drugs, including acyclovir, such systems are insufficiently explored and poorly studied. Therefore, this manuscript is a very important contribution to the development of mucoadhesive preparations with antiviral activity to protect women from the high incidence of HSV. The study is done very comprehensively with the application of several complementary techniques, from characterization of tablets of various compositions to the release of Acyclovir from them and interactions with mucin of solid polymers from animal tissue. I believe that this manuscript once published will become a key and very important reference regarding mucoadhesive characteristics of this important class system for controlled release drug. This is an interesting, complete and well designed paper, and I think, it deserves publication after a minor revision.

  1. I believe that the results section should begin with the characteristics of the substances, in this case tablets. Please in the results section before Infrared spectroscopy give the designation of the obtained tablets and a brief description of their receipt. This will help to better perceive the results, since the designation of tablets is given only at the end of the article in the methods.

Authors’ response 1: As the reviewer suggested, we have included in the revised manuscript a short paragraph at the beginning of the section 2 where we briefly describe the processing method for manufacturing the tablets. Therefore, the numbering in the results section has been suitably modified. On the other hand, in Methods section 3.2.1. Preparation of tablets and infrared spectroscopy has been modified in order not to be repetitive with the results section.

  1. Please transfer table 3 from the methods section to the results section.

Authors’ response 2: According to the reviewers suggestion, Table 3 has been moved to results and discussion section as Table 1 for the sake of clarity and the table numbering has been suitably modified in the revised manuscript.

  1. Reviewer’s comment: Please show clearly in the infrared spectroscopy image a range from 1000 to 500 nm. It is know that IR spectra of sulfated carbohydrates display absorption bands in the region of 800-850 nm, and axial sulfate at position 2 of a 3,6- anhydrogalactose residue has a specific absoption band at 805 nm characterizated of iota carrageenan. This area is poorly shown in Figure 1. For quantitative evaluation, it is better to obtain the spectrum of adsorption, but not transmission.

Authors’ response 3: The reviewer is right in the assertion that sulfated carbohydrates display absorption bands in the region of 800-850 nm, and axial sulfate at position 2 of a 3,6- anhydrogalactose residue has a specific absoption band at 805 nm characterizated of iota carrageenan. This is true in NIR spectroscopy where the spectra are collected in the near infrared light range comprised between 400 and 2500 nm (at maximum). We are collecting our spectra in the infrared range comprised between 600 and 4000 cm-1, that means 16667 to 2000 nm. Quite far away from the previously mentioned ranges. In the spectral range where we are collecting our spectra, we basically obtaining chemical information from the fundamental vibrational transitions. In contrast, the signal that was measured in NIR spectroscopy originates from overtones (multi-quanta transitions) and combinations of the fundamental vibrations.

Our choice of selection for FTIR spectroscopy is that in this wavelength range we obtain information of most of the components in the tablets and their interactions as well. In the case of NIR, we are always limited by the low sensitivity of the signal, so the interaction between components in the tablets could be mislead.

Reviewer 2 Report

Overall the manuscript is interesting, has good readability, and features essential sets of data for characterizing bioadhesive hydrogel systems.  The reviewer deem the manuscript publishable in Marine Drugs after clarifying the following points and relflect any clarifications in the revision.  No major issues.

  1. From the ATR-FTIR, the authors stated that ACV and the matrix are ‘compatible’. Does it mean that there are significant intermolecular interactions with any components of the gel materials?
  2. Figure 2: The y-axis is labeled as swelling ratio (%). However, strictly speaking, the gel mass change is caused by both swelling (mass gain) and erosion (mass loss) events. The reviewer suggests ‘(Relative) gel mass change (%)’ or equivalent. For the figure caption, the reviewer suggests ‘Swelling/Erosion profiles’ instead of ‘Swelling profiles’.  
  3. What are the advantages of this system compare to the bioadhesive liposomal vaginal DDS developed by Jalsenjak et al in 2005 (published in J. Controlled Rel. and Int. J. Pharm.)?
  4. The pore size distribution reversely correlates with ACV drug release rates in vitro. Although the authors provided plausible release mechanisms using mathematical models, the manuscript does not quite explain this counterintuitive observation, considering that ACV is not terribly hydrophobic and therefore the release may be facilitated with increasing pore volumes. Apparently the pore sizes within the gel matrices increased with the increasing amount of HPMC incorporated. Does it mean that HPMC content acts as a controlling factor for ACV release, perhaps through significant interactions between ACV and HPMC that sufficiently overcomes the increasing pore volumes as we move from I, I1H, I2H, H? 

Author Response

Response to Reviewer 2 Comments

Reviewer’s comment: Overall the manuscript is interesting, has good readability, and features essential sets of data for characterizing bioadhesive hydrogel systems. The reviewer deem the manuscript publishable in Marine Drugs after clarifying the following points and reflect any clarifications in the revision.  No major issues.

  1. From the ATR-FTIR, the authors stated that ACV and the matrix are ‘compatible’. Does it mean that there are significant intermolecular interactions with any components of the gel materials?

Authors’ response 1: The term compatibility here implies that there are no significant interactions between the matrix and the drug molecule and thus, they can be combined without risk of degradation of any of the components, neither the polymers nor the MgSt. We have clarified that in the revised manuscript.

  1. Reviewer’s comment: Figure 2: The y-axis is labeled as swelling ratio (%). However, strictly speaking, the gel mass change is caused by both swelling (mass gain) and erosion (mass loss) events. The reviewer suggests ‘(Relative) gel mass change (%)’ or equivalent. For the figure caption, the reviewer suggests ‘Swelling/Erosion profiles’ instead of ‘Swelling profiles’.  

Authors’ response 2: Y-axis in Figure 2 has been changed according to the Reviewer´s suggestion by (Relative) gel mass change (%) and in the revised manuscript. Also, the authors have changed Swelling profiles by Swelling/Erosion profiles in Figure 2 caption according to their comments.

Moreover, we have homogenized the text according to this new labeling and now, we refer in all the cases to the Swelling/erosion profiles instead of swelling profiles.    

  1. Reviewer’s comment: What are the advantages of this system compare to the bioadhesive liposomal vaginal DDS developed by Jalsenjak et al in 2005 (published in J. Controlled Rel. and Int. J. Pharm.)?

Authors’ response 3: In the first study (“Development and in vitro evaluation of a liposomal vaginal delivery system for acyclovir”), they only focused on the development and characterization of a liposomal system for vaginal administration of ACV. It is observed that they showed the results up to 24h of ACV release but it is not confirmed how long the formulation completely release of the drug, which limits us to compare it with our study, although when observing the % released at 24h, it is observed that our formulations (ACV-IH1 and ACV-IH2) have a better behaviour, since they release less amount of drug in that time. Another circumstance that should be analyzed is that in our study ex vivo mucoadhesion tests were performed, which demonstrated the mucoadhesive efficacy of the formulations for prolonged periods of time (up to 192h). Finally, another important aspect of our study is the application of iota-CG in the formulations, since its proven antiviral capacity against HVS and HPV, makes it an antiviral adjuvant to the formulation.

In the other study (“Characterization and in vitro evaluation of bioadhesive liposome gels for local therapy of vaginitis”), they developed liposomal systems with other drugs (Clotrimazole and Metronidazole), which they do not show the time total drug release, but at this time there was a release of more than 50% of the drugs, when evaluated in simulated vaginal fluid (SVF). They also demonstrated that the liposomes were more stable at pH 7.4, but when evaluated in SVF together with the hydrogel used as a vehicle, the stability of the liposomes decreased due to the presence of the components of the simulated vaginal fluid.

Therefore, it can be shown that the proposed formulations have better advantages over these systems, since they have been evaluated under conditions that simulate those of the human being (simulated vaginal fluid pH 4.2), which allowed obtaining a controlled release of ACV, at different times from 48 to 144h. In addition, the mucoadhesiveness of the formulations was evaluated, showing that three of the four formulations have prolonged residence time, allowing the complete release of the drug (ACV-IH1 and ACV-IH2). Consequently, we can mention that the tablets have a greater number of advantages over the semi-solid formulations, since these systems can present leaks, disorder, limitation in their application, inadequate doses, low residence time and perhaps these systems do not completely control the release of the drug. In conclusion, the versatility of liposomes such as flexibility for structure modification, adjustable characteristics and reliable protection of drug agents is unquestionable. Nevertheless, as we point out in our revised manuscript, liposomal vaginal drug delivery systems possesses similar limitations as the semi-solid systems as well as the easiness of degradation of the liposomes thus providing a fast release of the drug molecules. Tablets, on their side, solves some of the disadvantages of the above referred systems.

Thus, the authors have added a brief paragraph in the introduction where they describe the advantages of their formulations with respect to those studied by Jalsenjak et al in 2005, adding the bibliographic citations corresponding to these authors to the total references.

  1. Reviewer’s comment: The pore size distribution reversely correlates with ACV drug release rates in vitro. Although the authors provided plausible release mechanisms using mathematical models, the manuscript does not quite explain this counterintuitive observation, considering that ACV is not terribly hydrophobic and therefore the release may be facilitated with increasing pore volumes. Apparently the pore sizes within the gel matrices increased with the increasing amount of HPMC incorporated. Does it mean that HPMC content acts as a controlling factor for ACV release, perhaps through significant interactions between ACV and HPMC that sufficiently overcomes the increasing pore volumes as we move from I, I1H, I2H, H? 

Authors’ response 4: The average pore sizes of the swelling witnesses corresponding to the tablets containing I and HPMC are in between the pore sizes of the systems just containing I (small size) and the systems containing HPMC (bigger size). This fact demonstrates that the combination of I and HPMC in the I1H and I2H tablets lead to the formation of a mixed gel where HPMC and I chains are mixed together. In the combined tablets, the aqueous media uptake differs from the uptake in tablets formulated with just one polymer. This evidently implies that the gelling characteristics of I/HPMC tablets are intermediate from the gel created by I and HPMC independently. This modified swelling process determines the ACV release.

When the mixed I/HPMC tablets are immersed in the aqueous media, the formed gels possess an intermediate consistency in between the gels formed just by I and the gels with HPMC. In the case of I, the consistency is quite low and they are consequently eroded whereas in the case of HPMC, possessing a high consistency, wetting is somehow hindered and even the core of the tablet remains unwetted. That is the main reason why ACV release from I/HPMC tablets is controlled but occurs faster than in the case of HPMC tablets.

The text describing this behaviour in the manuscript states as:

“For batches ACV-IH1 and ACV-IH2, the total release of ACV occurs at 96 h, representing an intermediate release compared to the other formulations. These formulations control the release of the drug more effectively than ACV-I, thanks to the consistency and volume of the gel formed by this combination. This is supported by the swelling profiles, since these mixtures have the highest swelling rate of all the tablets, allowing a faster release compared to ACV-H. There are no significant differences in the release profiles of ACV-IH1 and ACV-IH2, regardless of the amount of HPMC or iota-CG. This confirms that the iota-CG/HPMC combination produces a mixed gel that can maintain its properties despite changes in the proportion of the two polymers, already described in other studies when HPMC is mixed with chitosan or kappa-carrageenan [13,15,62,63].”